# Unveiling the Role of Zoos in Smart Cities: A Quantitative Analysis of the Degree of Smartness in Kyoto City Zoo

Yuxuan Lin [1], Ruochen Yang [1,*], Ryosuke Shimoda [1], Zheng Xian [2] and Shuhao Liu [1]

1    Graduate School of Horticulture, Chiba University, Matsudo 271-8510, Chiba, Japan; 22hd0401@student.gs.chiba-u.jp (Y.L.); 21hd0503@student.gs.chiba-u.jp (S.L.)
2    College of Landscape Architecture, Nanjing Forestry University, Nanjing 210037, China; xiankoumuzi.sean@gmail.com
*    Correspondence: 21hd0501@student.gs.chiba-u.jp

**Abstract:** The rapid pace of urbanization and the emergence of social challenges, including an aging population and increased labor costs resulting from the COVID-19 pandemic, have underscored the urgency to explore smart city solutions. Within these technologically advanced urban environments, zoos have assumed a pivotal role that extends beyond their recreational functions. They face labor cost challenges and ecological considerations while actively contributing to wildlife conservation, environmental education, and scientific research. Zoos foster a connection with nature, promote biodiversity awareness, and offer a valuable space for citizens, thereby directly supporting the pillars of sustainability, public engagement, and technological innovation in smart cities. This study employs a quantitative analysis to assess the alignment between smart projects and the distinctive characteristics of Kyoto Zoo. Through questionnaires, we collected feedback on performance and importance, and subsequently employed the analytic hierarchy process and the fuzzy integrated evaluation method to obtain quantitative results. The findings reveal the high level of intelligence exhibited by Kyoto Zoo, and the analysis provides insightful guidance that can be applied to other urban facilities. At the same time, we compared Kyoto Zoo with Ueno Zoo to see the difference in intellectualization achievements in different contexts in terms of data and systems.

**Keywords:** smart zoo; smart city; intellectualization; fuzzy comprehensive evaluation method (FCEM); importance–performance analysis (IPA); smart zoo system of Japan (SZSOJ); Kyoto City Zoo





## 1. Introduction

### 1.1. Background

The rapid expansion of urban areas, coupled with advances in technology and the need to improve citizens' living conditions and well-being, has placed greater emphasis on the role of landscapes in the city. In response to these challenges, the "Smart City" concept has emerged, drawing on the notion of a "Smart Earth" first introduced by IBM in a thematic report in 2008 [1]. A smart city is a modernized urban environment that leverages diverse electronic methods and sensors to collect specific data, intending to manage assets, resources, and services effectively and ultimately enhance overall city operations [2]. Integrating information and communication technology (ICT) and Internet of Things (IoT) technologies into smart cities has enabled greater information transparency and digitalization of city life, empowering citizens with the tools and data they need to make informed choices on a daily basis.

The concept of "Smart" (Japanese for 'sumāto-ka') has garnered significant attention in urban development and intellectualization, as reflected by its widespread adoption across various industries. Key terms such as ICT, IoT, artificial intelligence (AI), and 5G are now firmly entrenched in the public consciousness. The growing prevalence of "Smart Cities" necessitates using the IoT as an information network platform, enabling the efficient

collection and processing of big data. The concept of smart cities goes beyond traditional urban development, aiming to optimize city operations and enhance the quality of life for citizens. By leveraging information and communication technology (ICT) and the Internet of Things (IoT), smart cities enable effective management of assets, resources, and services. Integrating ICT and IoT technologies empowers citizens with real-time data and tools, fostering information transparency and digitalization in various aspects of urban life. These advancements provide the foundation for a smarter and more efficient urban environment.

With urban development on the rise, there has been a surge in the construction of "Smart Parks", such as Haidian Park [3], Longhu G-PARK Science Park [4] in Beijing, China, Xiangmi Park [5] in Shenzhen, China, Arashiyama Park (Nakanoshima area) [6], and The Keihanna Commemorative Park [7] in Kyoto, Japan, and the Palace Site Historical Park [8] in Nara, Japan. These parks represent an innovative approach to providing citizens with better green spaces. Within this context, our study focuses on a specific type of park, namely zoos. Zoos play a vital role in developing smart cities by serving as integral components of urban landscapes. These institutions contribute to cities' overall well-being and sustainability by providing green spaces, wildlife conservation efforts, and opportunities for education and research. In the context of smart cities, zoos act as catalysts for sustainable development and success, aligning with the core principles and objectives of these technologically advanced urban environments. The more targeted visitor traffic and richer ecological environments in zoos make their intellectualization more impactful and meaningful.

Beyond their role as recreational spaces, zoos fulfill critical functions such as wildlife conservation, environmental education, and scientific research. These activities directly contribute to the sustainable development, public engagement, and technological innovation aspects of smart cities. Zoos not only provide citizens with opportunities to connect with nature but also serve as platforms for raising awareness about biodiversity and environmental sustainability.

The COVID-19 pandemic has significantly impacted zoos worldwide, with many facing operational and financial difficulties. The decrease in visitor numbers, which is one of the main sources of revenue for zoos, has severely impacted their operations. In addition, the increased costs of maintaining the animals and providing them with food and other necessities have also contributed to the financial difficulties that zoos face. To cope with these challenges, zoos have implemented cost-cutting measures, reducing staff, animal collections, and conservation programs.

In Japan, for example, in 2020, feed costs at Tobu Zoological Park [9] increased by ~5–6%. In a 2021 survey conducted by NHK, 97% of zoos in Japan said they had closed temporarily during the prior year [9]. Since then, admission revenues in tourism in Japan have decreased to a staggering number due to a sharp decline in inbound visitors from overseas [10]. It can be seen from this that supporting and sustaining zoos during crises is crucial, given their significant contributions to animal conservation, education, and research. It is imperative to find ways to overcome these challenges and ensure the long-term viability of zoos in the context of smart cities.

*1.2. Cases and Situation*

Innovative smartening projects have been implemented in Japan to mitigate the negative impact of the COVID-19 pandemic on zoos. For instance, KDDI, a Japanese company, launched the "one zoo" online platform, which featured prominent zoos such as the Asahiyama Zoo and the Tennoji Zoo [11]. The platform allowed users to observe animals in real time and make donations to animal protection associations through membership purchases. Additionally, the platform rewarded users with zoo tickets or souvenirs. However, despite the developers' efforts to enhance the zoo tour experience, the project was discontinued on 31 May 2022 [11] due to a lack of online activity. The developers had not considered user feedback on each smartening project promptly and lacked objective analysis, leading to the project's failure.

Another example is Tokyo Zoonet's online platform [12], Tokyo Zoovie, which comprises four members of the Tokyo Zoological Park Society (Tokyo Dobutsuen Kyokai): Ueno Zoological Gardens, Tama Zoological Park, Tokyo Sea Life Park, and Inokashira Park Zoo. The platform provides visitors with a guided tour of the four zoos using an animal map and 3D models, and VR tours are also available. In addition, Ueno Zoo is part of the Tokyo Metropolitan Park Association, and it offers smart functions in the Tokyo Parks Navi platform, such as the ability to collect stamps, look up tour routes, blogs, and automatic tour recommendations, making it very user-friendly.

The development of smart platforms for zoos, such as "one zoo" and Tokyo Zoonet, highlights the increasing utility of intellectualization in addressing the operational and financial difficulties these institutions face. However, it is crucial to objectively assess the practicality and effectiveness of these smart functions and determine whether there is actual demand from visitors for such features. To this end, this study aims to model and analyze these issues quantitatively, enabling zoo managers to make informed decisions regarding the zoo's development, identify potential cost savings, and gain insight into visitor needs and preferences compared to the wider market. By providing an objective and data-driven analysis of the efficacy of smart functions in zoos, this research will contribute to these vital institutions' sustainable development and success.

The current state of Japanese smart zoos is in a preliminary phase, necessitating a standardized and objective set of regulations to identify good and bad smart implementations. Nevertheless, at the current stage, most smart projects are focused on multimedia functions to enhance the visitor experience. There are relatively few projects centered on big data and ecological conservation. Thus, the judgment criterion will focus on visitor feedback rather than efficacy values. The data collection component of this study will take the form of a questionnaire, asking respondents to rate the importance and performance of each smart item on a scale from 1 to 5. AHP (analytical hierarchy process) weights will be calculated based on this questionnaire data, and FCEM (fuzzy comprehensive evaluation method) will be employed to obtain numerical results for the objective indicators of intellectualization. In addition, IPA (importance–performance analysis) will be utilized to evaluate each smart project, assess its current development status, and obtain opinions. Through this study, zoo managers can identify the appropriate direction for zoo development, achieve significant cost savings, determine visitor needs and preferences, and compare their zoo with the broader market.

As demonstrated in our previous research, we have already conducted a comprehensive examination of Ueno Zoo in Tokyo using the above-mentioned methodology [13]. Moreover, for the current investigation, our focus will shift to Kyoto Zoo in Kyoto City, a highly illustrative metropolis that has experienced a fiscal crisis in the past ten years [14], prompting an extensive effort to revitalize its economic landscape through a multifaceted smart city plan. Kyoto Zoo is an ideal site for our research because of this complex milieu. Furthermore, our investigation aims to explore the divergences between the intellectualization of zoos as a general practice and the unique challenges and opportunities that arise from zoo development within the comprehensive framework of a smart city.

### 1.3. Literature Review

The current discourse in Japan within the academic community has shifted toward embracing the notion of smart zoos. However, it is important to note that the term commonly used in Japan is "Intellectualization of zoos" (or Dōbutsuen no sumāto-ka), which is often regarded as an integral component of the broader smart parks concept.

"SMART PARK: A TOOLKIT", from the Luskin School of Public Affairs, UCLA [15], provides a comprehensive understanding of the concept of smart parks, laying out a framework for evaluating such parks based on their spatial characteristics from the perspective of designers, park managers, and advocates. While this model offers a satisfactory level of specificity in defining various program parameters, the missing objective data evaluation system remains a critical gap. Similarly, "Research on the Construction

Framework of Smart Park: A Case Study of Intelligent Renovation of Beijing Haidian Park" offers a systematic approach to evaluating smart parks based on their functions [3]. However, the study does not include a comprehensive survey of tourists' emotions and objective data, limiting its applicability. To address this gap, the article "How smart is your tourist attraction? Measuring tourist preferences of smart tourism attractions via an FCEM-AHP and IPA approach" [16] adopts a pioneering approach to incorporate FCEM–AHP and IPA methods into analyzing the weighting of parks and tourism preferences. The study leverages a questionnaire to collect data and uses AHP to determine weight sets, while a fuzzy comprehensive evaluation approach is applied to derive the strengths and weaknesses of the park. Although the study model provides a comprehensive framework, it has several limitations, including the lack of clear project descriptions and illustrations in the questionnaire, resulting in limited understanding among interviewees. Additionally, many of the projects in the study require re-exploration due to changes over the past few years.

Research on smart parks has recently entered an initial stage, with the establishment of frameworks for evaluating spatial characteristics and functional aspects. However, a critical gap needs to be addressed regarding objective data evaluation systems and comprehensive surveys of tourist feedback data. Previous studies have made notable contributions by adopting innovative approaches like FCEM–AHP and IPA methods to analyze park weighting, tourism preferences, and strengths and weaknesses. These studies lay a solid foundation for further research on smart zoos, particularly focusing on the Kyoto Zoo within the context of Kyoto Smart City.

### 1.4. Research Purpose and Significance

In another prior investigation, "Impact of Intellectualization of a Zoo through a FCEM-AHP and IPA Approach", the study pursued a methodical evaluation of the intellectualization process of Ueno Zoo [13]. The outcome revealed that Ueno Zoo is still in the nascent stage of intellectualization, with several components requiring further development for visitors to have an immersive tourist experience. Therefore, there is a pressing need to enhance the intellectualization and user-friendliness of the Tokyo zoos to create a more comprehensive and satisfactory tourist experience. Previous studies have provided a solid foundation for further research on smart zoos, with particular attention to Kyoto Zoo, utilizing the FCEM and IPA methodologies. Moreover, employing the same analytical framework would facilitate the comparative analysis of the degree of smartness and development orientation between Kyoto Zoo and Ueno Zoo. By employing the FCEM and IPA methodologies, the present study aims to quantitatively evaluate the intellectualization of Kyoto Zoo and compare it with Ueno Zoo, utilizing a consistent analytical framework. The ultimate goal is to enhance the intellectualization and user-friendliness of zoos in Japan, providing a more comprehensive and satisfactory tourist experience.

## 2. Materials and Methods

### 2.1. Study Area

The selection of Kyoto Zoo as the study site was deliberate and based on several reasons. Firstly, it is the second-oldest zoo in Japan, after Ueno Zoo, and has a rich history and heritage. Secondly, Kyoto Zoo is a non-commercial entity that espouses humanistic values and promotes peace. In 1941, during the war, many animals at the zoo perished through a large-scale animal slaughter. Since 1942, the zoo has held memorial services almost every autumn to express gratitude and reinforce the importance of life [17]. As of June 2019, Kyoto Zoo is home to 570 animals of 123 species, comprising mammals, birds, reptiles, amphibians, and fish [18]. Hence, Kyoto Zoo is where visitors can appreciate animals, ponder their living conditions, experience life through animal interaction, and gain insights into human–nature relationships. It embodies a part of Kyoto's culture and revered traditions and underscores the importance of peaceful coexistence between animals and humans. Thirdly, although the zoo is not located in the city center, it is situated in the Okazaki Area of Kyoto, which is surrounded by popular tourist destinations such

as the Kyoto City Kyocera Museum of Art, Okazaki Park, and the Heanjingu Shrine, thereby ensuring a steady flow of visitors and a conducive operating environment [19]. During the financial crisis faced by Kyoto City, Kyoto Zoo appealed for assistance via SNS platforms, seeking support from local shops and donations from the community to provide the animals a chance to survive [20]. Notably, a local pickle store donated radish roots and leaves not commonly consumed by humans to serve as animal food. This act served as an example of how intellectualization can contribute to regional collaboration and promote or influence certain sustainable development goals (SDGs).

### 2.2. Identifying Evaluation Items of the Smart Zoo System of Japan (SZSOJ)

The research process of this study is shown in Figure 1.

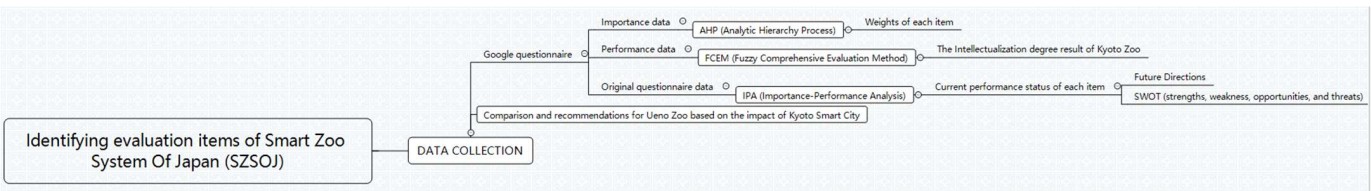

**Figure 1.** Research process.

The present study seeks to explore the unique application of the concept of intellectualization in Japanese zoos, which is closely intertwined with the urban lifestyle of Japan. To achieve this aim, we draw upon ongoing projects at Kyoto Zoo, which has been observed to have a wide range of QR codes, making it a noteworthy feature for our primary classification. Furthermore, the zoo's official ecological sustainability plan identifies the ecosystem as another primary classification item. Our survey revealed that the mobile application in Kyoto Zoo has been discontinued. As a result, we have identified four primary classification items: QR code information function, ecology system, functions within the zoo, and official website function. The 26 secondary classification items are derived from these four primary categories. A summary of the concept definitions of these items is presented in Table 1.

**Table 1.** Original evaluation items of SZSOJ.

| Primary Classification | No. | Secondary Classification | Factor Description |
|---|---|---|---|
| QR code information function | 1 | Plant QR code information | QR codes on plants can provide information about their scientific name, habitat, and conservation status, allowing visitors to learn more about the flora in the zoo. |
| | 2 | Animal QR code information | QR codes on animal enclosures can provide information about the animal's species information, habitat, diet, and behavior, enabling visitors to learn about the zoo's fauna. |
| | 3 | Media report QR code information | QR codes on media reports can provide visitors with additional information about the animals and plants in the zoo, as well as the zoo's history, mission, and ongoing projects. |
| | 4 | Questionnaire research QR code information | QR codes on questionnaires can enable visitors to provide feedback and suggestions to the zoo, which can be used to improve the visitor experience and animal welfare. |
| | 5 | Artwork QR code information | QR codes on artworks in the zoo can provide information about the artists, materials, and themes of the artworks, enhancing visitors' appreciation of the zoo's artistic and cultural value. |
| | 6 | Academic research QR code information | QR codes about academic research can provide visitors access to scientific publications and reports on the zoo's conservation and animal welfare efforts. |
| | 7 | Event QR code information | QR codes about events can provide visitors with schedules, maps, and descriptions of the activities and performances taking place in the zoo. |

**Table 1.** *Cont.*

| Primary Classification | No. | Secondary Classification | Factor Description |
|---|---|---|---|
| | 8 | Animal education science videos QR code information | QR codes for science videos can provide visitors with educational and entertaining content about animal behavior, ecology, and conservation, enhancing their understanding of the natural world. |
| | 9 | Regional activities QR code information | QR codes about regional activities can inform visitors about cultural and recreational activities in the zoo's surrounding area, encouraging them to explore the local community. |
| | 10 | Animal education science live QR code information | QR codes about live animal education events can provide visitors access to real-time animal behavior and conservation education, promoting a deeper understanding and appreciation of the zoo's mission. |
| | 11 | Animal protection organization QR code information | QR codes about animal protection organizations can inform visitors about partner organizations and their efforts to conserve and protect endangered species worldwide. |
| | 12 | Ecological cycle systems | Ecological cycle systems in the zoo can sustainably manage waste, recycle resources, and maintain a healthy environment for animals and plants. |
| | 13 | Environmental sensors | Environmental sensors can monitor the zoo's temperature, humidity, air quality, and other environmental factors, providing data for environmental management and animal welfare. |
| | 14 | Automatic watering | Automatic watering systems can provide plants with appropriate amounts of water, thereby reducing water waste and ensuring plant health in the zoo. |
| Ecology system | 15 | Eco-energy (solar power) | Solar power can generate clean energy for the zoo, reducing its carbon footprint and promoting sustainable energy use. |
| | 16 | Ecological energy use information | Information about ecological energy use in the zoo can educate visitors about the zoo's efforts to reduce energy consumption, promote renewable energy, and protect the environment. |
| | 17 | Free WIFI | Free WIFI in the zoo can provide visitors access to online resources and enhance their overall experience. |
| | 18 | Electronic ticketing system | Electronic ticketing systems can streamline ticket purchasing and reduce wait times for visitors, improving their overall experience in the zoo. |
| | 19 | Interactive animal education | Interactive animal education can provide visitors with engaging and educational experiences, such as allowing them to interact with animals through devices or providing real-time feedback on animal behavior and health, promoting a deeper understanding and appreciation of the natural world. |
| Functions within the zoo | 20 | Animal state observation | Animal state observation can monitor animal behavior and health, enabling the zoo to provide appropriate care and promote animal welfare. |
| | 21 | Animal status detection (camera) | Animal status detection cameras can detect and monitor animal behavior and health, providing data for animal welfare management and research. |
| | 22 | Electronic information screen | Electronic information screens can provide visitors with maps, schedules, and other relevant information about the zoo, enhancing their overall experience. |
| | 23 | Smart souvenir vending (photos) | Smart souvenir vending machines can provide visitors with customized photo souvenirs, enhancing their zoo memories and promoting sustainable souvenir production. |
| | 24 | Official website function | The zoo's official website provides visitors with comprehensive information about the zoo's animals, exhibits, events, and services. |
| Official website function | 25 | Tourism SNS | The zoo uses social media platforms such as Facebook, Instagram, and Twitter to promote tourism activities and interact with visitors. |
| | 26 | Digital map | The digital map of the zoo is accessible on mobile devices. It provides visitors real-time information about exhibits, events, and animal locations, facilitating navigation and enhancing the visitor experience. |

### 2.3. Data Collection

This study gathered data from 117 highly qualified graduate students in landscape architecture enrolled at prestigious universities in Kyoto and Chiba. To ensure the veracity and credibility of the collected data, respondents were required to log in to their personal accounts before answering the Google questionnaire. Additionally, participants confirmed that they had experienced the Kyoto Zoo as a tourist, thus providing reliable insights into the smart zoo experience. Due to their academic backgrounds, the respondents could evaluate the smart zoo experience from a research-based perspective, while the completed tourist experience guaranteed the validity of the questionnaire. The questionnaire was designed with two levels of indicators (Level 1 and 2), and it included items that were assessed for their importance and performance on a scale of 1–5. The importance assessment scale ranged from 1 (not at all important) to 5 (very important), whereas the performance assessment scale ranged from 1 (very poor) to 5 (very good). The inclusion of graphical descriptions in each item aimed to prevent misidentification. The reliability of the questionnaire was also tested to ensure its quality. For the full list of questionnaire items, please refer to Supplementary File S1.

In order to derive meaningful insights from the collected data, the study utilized a two-stage process. The importance rankings obtained from the survey results were utilized as objective data references in the first stage. To determine the weightage of each item, the study applied the AHP. The AHP-derived weights were then used in the FCEM. This method integrates the fuzzy theory, a widely recognized method for decision making in complex situations, and the analytic hierarchy process to evaluate complex systems. The FCEM was utilized to obtain the zoo's current results for construction effectiveness.

In the second stage, the original 1–5 rating data obtained from the questionnaire were retained. The study employed IPA testing to assess the overall intellectualization construction degree and each specific item in the zoo. IPA is a widely used method for evaluating the performance of a system or product by examining the relationship between importance and performance. The results obtained from the IPA testing were then used to guide future zoo development, providing valuable insights that could be used to enhance the visitor experience and improve the zoo's overall effectiveness.

### 2.4. AHP (Analytic Hierarchy Process)

The analytic hierarchy process (AHP) is an essential tool for this study due to its rigorous and systematic approach to decision making. Developed by Thomas L. Saaty in the mid-1970s [21], the AHP combines qualitative and quantitative analyses to quantify group decisions and priorities. By breaking down complex problems into hierarchical structures and using pair-wise comparisons, the AHP determines the relative importance and weight of criteria and alternatives [22]. This allows decision makers to make well-informed and transparent choices based on thorough analysis. Therefore, in our study, we adopted the AHP as a recognized method for systematically and hierarchically quantifying group decisions and weights. We used a pair-wise comparison of the weights of each item to assess the relative importance of different criteria within each item. To ensure the accuracy of the pair-wise comparison process, importance rankings were collected from the questionnaire, and the resulting data were transformed into percentages on a scale of 1–9. These percentages were then used to judge the relative importance of pair-wise comparisons among all items. The rankings of relative importance, as shown in Table 2, were obtained from this process.

**Table 2.** Scales of relative importance.

| Scales of Relative Importance | Meaning |
| --- | --- |
| 1 | Equally important |
| 3 | Slightly important |
| 5 | Quite important |

**Table 2.** *Cont.*

| Scales of Relative Importance | Meaning |
| :---: | :---: |
| 7 | Obviously important |
| 9 | Absolutely important |
| 2, 4, 6, 8 | Intermediate scales |
| 1/3 | Slightly unimportant |
| 1/5 | Quite unimportant |
| 1/7 | Obviously unimportant |
| 1/9 | Unimportant |
| 1/2, 1/4, 1/6, 1/8 | Intermediate scales |

The vector U will also define each evaluated item set.
The classification is defined as follows:

$$U = \{U_1, U_2, U_3, U_4\}$$

$$U_1 = \{U_{11}, U_{12}, U_{13}, U_{14}, U_{15}, U_{16}, U_{17}, U_{18}, U_{19}, U_{110}, U_{111}\}$$

$$U_2 = \{U_{21}, U_{22}, U_{23}, U_{24}, U_{25}\}$$

$$U_3 = \{U_{31}, U_{32}, U_{33}, U_{34}, U_{35}, U_{36}, U_{37}\}$$

$$U_4 = \{U_{41}, U_{42}, U_{43}\}$$

where U represents the total set of all items, $U_1$ to $U_4$ correspond to the four first-level categorization items in an order and subsets $U_{11}$ to $U_{43}$ of $U_1$ to $U_4$ correspond to the second-level categorization items contained under each first-level categorization item.

The AHP method analyzes designated items based on their importance ranking and then constructs a judgment matrix. The maximum eigenvalue of the judgment matrix is calculated, and the resulting eigenvector is considered the evaluation weight vector A. However, a consistency test is performed to ensure the objectivity and rationality of the judgment. This is because the AHP method is prone to inconsistencies in the judgment matrix when respondents are asked to compare the importance of multiple criteria. Therefore, a consistency ratio (CR) is calculated to determine the degree of inconsistency in the judgment matrix.

In the entirety of the computation, the deviation consistency index of the judgment matrix is represented by *CI*, which is calculated as $CI = \frac{(\lambda - n)}{(n-1)}$. A higher value of *CI* indicates poor consistency of the judgment matrix, whereas a *CI* value of 0 represents the complete character of the matrix. The consistency ratio, denoted as CR, is calculated using the formula $CR = \frac{CI}{RI}$, where *RI* represents the average random consistency index. When CR < 0.1, the consistency of the judgment matrix can be considered acceptable.

*2.5. FCEM (Fuzzy Comprehensive Evaluation Method)*

The fuzzy comprehensive evaluation method (FCEM) is needed for this study due to its ability to handle uncertainty and imprecise information. Based on the fuzzy set theory pioneered by Lotfi Zadeh [23], the FCEM allows for representing and manipulating fuzzy and uncertain data. With its application in various fields, FCEM enables the conversion of qualitative and uncertain assessments into quantitative measurements [24]. In this study, where perceptions of the concepts of "Smart" for visitors are inherently vague, FCEM is employed to analyze and evaluate the effectiveness of smart construction in the zoo. By utilizing FCEM, the study aims to provide a comprehensive assessment considering multiple factors and constraints.

The FCEM calculation process is carried out in two steps using MATLAB. The first step involves establishing the fuzzy judgment matrix. The degree of membership of the Item Set Rm can be defined as follows:

$$R_m = \begin{bmatrix} R_{m1a} & R_{m1b} & \cdots & R_{m1e} \\ R_{m2a} & R_{m2b} & \cdots & R_{m2e} \\ \vdots & \vdots & \ddots & \vdots \\ R_{mna} & R_{mnb} & \cdots & R_{mne} \end{bmatrix}$$

The weighting of Item Set A of the first classification calculated by AHP can be defined as:

$$A = \begin{bmatrix} A_1 & A_2 & A_3 & A_4 \end{bmatrix}$$

The weighting of Item Set Wm of the secondary classification calculated by AHP can be defined as:

$$W_m = \begin{bmatrix} W_{m1} & W_{m2} & \cdots & W_{mn} \end{bmatrix}$$

As mentioned above, the symbol "$m$" signifies the primary classification category, while "$n$" denotes the number of sub-classification items. Moreover, the symbols "$a$–$e$" correspond to the five-point rating system, ranging from 1 to 5. By using this method, the degree of membership of the Item Set Rm can be established. The collected raw data from the questionnaire are then transformed into the "R" matrix, which is utilized to construct the fuzzy judgment matrix.

The second step is to use the established matrix for the fuzzy comprehensive evaluation calculation as follows:

$$C_1 = W_1 \times R_1$$

$$C_2 = W_2 \times R_2$$

$$C_3 = W_3 \times R_3$$

$$C_4 = W_4 \times R_4$$

$$B = A \times \begin{bmatrix} C_1 & C_2 & C_3 & C_4 \end{bmatrix} = \begin{bmatrix} b_1 & b_2 & b_3 & b_4 & b_5 \end{bmatrix}$$

The term "$bi$" value refers to the degree of membership of the evaluated item to each evaluation criterion, which is determined based on the evaluation statement (e.g., "excellent", "good", "moderate", "fair", and "poor") corresponding to the ranking system. The "$bi$" value is obtained by performing the fuzzy calculation based on the degree of membership between the evaluation statement and the evaluated item. The highest value obtained from this calculation represents the intellectualization result of Kyoto Zoo, indicating the zoo's level of intelligence and smartness in terms of its facilities, exhibits, and services.

*2.6. IPA*

The importance–performance analysis (IPA) is a widely used method for evaluating customer satisfaction by measuring the gaps between customer expectations and actual perceptions [25]. Utilizing a four-quadrant diagram, this method can swiftly identify the areas requiring attention, prioritize each demand indicator, and formulate a sound implementation plan. The IPA method has proven to be an effective and straightforward approach for measuring customer satisfaction and improving the quality of service [26]. Its ease of use and practicality make it a valuable tool for businesses seeking to enhance customer satisfaction and stay ahead of the competition.

The mean value was computed for each item in the original questionnaire to perform IPA, and the resulting means for overall performance and importance were utilized as quadrant dividers. Figure 2 illustrates the chart that determines the position and stage of each item.

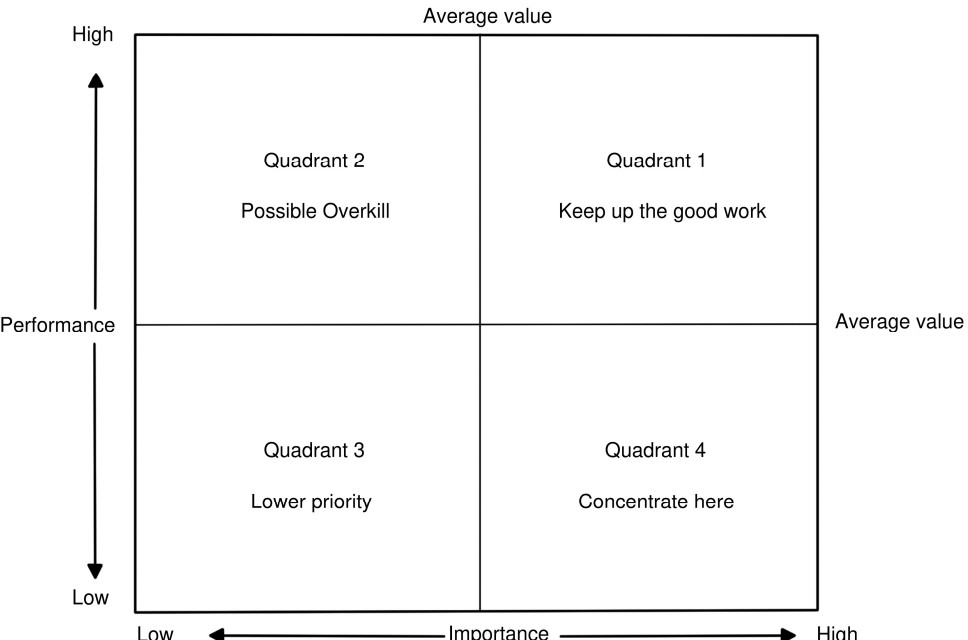

**Figure 2.** IPA matrix.

## 3. Results

### 3.1. Results of the AHP

The questionnaires demonstrated excellent recovery rates, and their reliability was assessed with values above 0.9. Additionally, validity was tested using the Kaiser–Meyer–Olkin measure of sampling adequacy with values greater than 0.5 and significant values less than 0.05. The detailed results are presented in Tables 3 and 4.

**Table 3.** Questionnaire reliability test.

|  | **Alpha** | **Number** |
| --- | --- | --- |
| **I and P** | 0.944 | 52 |
| **P** | 0.930 | 26 |
| **I** | 0.903 | 26 |

**Table 4.** Questionnaire validity test.

| **KMO and Bartlett's Test** | | | |
| --- | --- | --- | --- |
| **Kaiser–Meyer–Olkin Measure of Sampling Adequacy** | | 0.815 | >0.5 |
| **Bartlett's Test of Sphericity** | Approx Chi-Square | 2969.662 | |
| | df | 1326 | |
| | Sig. | <0.001 | <0.005 |

The relative importance of the questionnaire and the corresponding factors are presented in Table 5, with the AHP scores ranging from 1 to 9, reflecting the pair-wise comparisons. The AHP scores were derived from the participants' relative judgments percentage and indicated the priority and significance of each item in the evaluation process.

**Table 5.** AHP importance scoring correspondence table.

| Primary Classification | Result of Importance | Factor Defined | AHP Score | Secondary Classification | Result of Importance | | | | | Factor Defined | AHP Score |
|---|---|---|---|---|---|---|---|---|---|---|---|
| | | | | | 1 | 2 | 3 | 4 | 5 | | |
| QR code information function | 29.06% | $U_1$ | 9 | Plant QR code information | 8.55% | 11.97% | 19.66% | 35.04% | 24.79% | $U_{11}$ | 7 |
| | | | | Animal QR code information | 13.68% | 13.68% | 19.66% | 27.35% | 25.64% | $U_{12}$ | 9 |
| | | | | Media report QR code information | 14.53% | 16.24% | 23.08% | 20.51% | 25.64% | $U_{13}$ | 9 |
| | | | | Questionnaire research QR code information | 15.38% | 11.97% | 19.66% | 27.35% | 25.64% | $U_{14}$ | 1 |
| | | | | Artwork QR code information | 10.26% | 13.68% | 20.51% | 33.33% | 22.22% | $U_{15}$ | 9 |
| | | | | Academic research QR code information | 21.37% | 7.69% | 20.51% | 23.93% | 26.50% | $U_{16}$ | 9 |
| | | | | Event QR code information | 16.24% | 14.53% | 15.38% | 31.62% | 22.22% | $U_{17}$ | 8 |
| | | | | Animal education science videos QR code information | 12.82% | 14.53% | 25.64% | 23.93% | 23.08% | $U_{18}$ | 7 |
| | | | | Regional activities QR code information | 14.53% | 9.40% | 21.37% | 33.33% | 21.37% | $U_{19}$ | 3 |
| | | | | Animal education science live QR code information | 11.97% | 19.66% | 14.53% | 28.21% | 25.64% | $U_{110}$ | 5 |
| | | | | Animal protection organization QR code information | 11.97% | 11.11% | 19.66% | 37.61% | 19.66% | $U_{111}$ | 5 |
| Ecology System | 23.93% | $U_2$ | 5 | Ecological cycle systems | 15.38% | 10.26% | 26.50% | 25.64% | 22.22% | $U_{21}$ | 7 |
| | | | | Environmental sensors | 14.53% | 14.53% | 19.66% | 28.21% | 23.08% | $U_{22}$ | 1 |
| | | | | Automatic watering | 17.09% | 10.26% | 22.22% | 25.64% | 24.79% | $U_{23}$ | 9 |
| | | | | Eco-energy (solar power) | 13.68% | 9.40% | 21.37% | 29.91% | 25.64% | $U_{24}$ | 5 |
| | | | | Ecological energy use information | 12.82% | 13.68% | 17.95% | 27.35% | 28.21% | $U_{25}$ | 3 |
| Functions within the zoo | 18.80% | $U_3$ | 1 | Free WIFI | 9.40% | 13.68% | 15.38% | 29.06% | 32.48% | $U_{31}$ | 4 |
| | | | | Electronic ticketing system | 5.13% | 12.82% | 19.66% | 29.91% | 32.48% | $U_{32}$ | 6 |
| | | | | Interactive animal education | 11.11% | 12.82% | 27.35% | 20.51% | 28.21% | $U_{33}$ | 6 |
| | | | | Animal state observation | 9.40% | 11.97% | 20.51% | 25.64% | 32.48% | $U_{34}$ | 4 |
| | | | | Animal status detection (camera) | 13.68% | 12.82% | 19.66% | 24.79% | 29.06% | $U_{35}$ | 8 |
| | | | | Electronic information screen | 6.84% | 11.97% | 24.79% | 22.22% | 34.19% | $U_{36}$ | 9 |
| | | | | Smart souvenir vending (photos) | 13.68% | 10.26% | 23.08% | 33.33% | 19.66% | $U_{37}$ | 1 |
| Official website function | 28.21% | $U_4$ | 8 | Official website function | 11.11% | 11.11% | 13.68% | 33.33% | 30.77% | $U_{41}$ | 7 |
| | | | | Tourism SNS | 10.26% | 17.09% | 20.51% | 28.21% | 23.93% | $U_{42}$ | 1 |
| | | | | Digital map | 10.26% | 14.53% | 21.37% | 22.22% | 31.62% | $U_{43}$ | 9 |

The AHP method involves a systematic and pair-wise comparison of all items based on their relative importance, leading to a judgment matrix for each evaluation factor. As presented in Table 6, the judgment matrix for the first-level evaluation factors of SZ-SOJ has been established using the AHP method. Moreover, Tables 7–10 display the judgment matrices for the second-level evaluation factors. The consistency of all matrices has been evaluated, and the results indicate the accuracy and validity of the AHP analysis in this study.

**Table 6.** Judgment matrix of SZSOJ's first-level evaluation factors.

| Factor | $U_1$ | $U_2$ | $U_3$ | $U_4$ | Eigenvector | Weight (%) |
|---|---|---|---|---|---|---|
| $U_1$ | 1 | 5 | 9 | 2 | 3.08 | 51.192 |
| $U_2$ | 0.2 | 1 | 5 | 0.25 | 0.707 | 11.752 |
| $U_3$ | 0.111 | 0.2 | 1 | 0.125 | 0.23 | 3.816 |
| $U_4$ | 0.5 | 4 | 8 | 1 | 2 | 33.241 |

$\lambda_{max}$ = 4.155, CI = 0.051, RI = 0.882, CR = 0.058 < 0.10.

**Table 7.** Judgment matrix of SZSOJ's second-level evaluation factors ($U_1$).

| Factor | $U_{11}$ | $U_{12}$ | $U_{13}$ | $U_{14}$ | $U_{15}$ | $U_{16}$ | $U_{17}$ | $U_{18}$ | $U_{19}$ | $U_{110}$ | $U_{111}$ | Eigenvector | Weight (%) |
|---|---|---|---|---|---|---|---|---|---|---|---|---|---|
| $U_{11}$ | 1 | 0.333 | 0.333 | 7 | 0.333 | 0.333 | 0.5 | 1 | 5 | 3 | 3 | 1.062 | 6.917 |
| $U_{12}$ | 3 | 1 | 1 | 9 | 1 | 1 | 2 | 3 | 7 | 5 | 5 | 2.54 | 16.535 |
| $U_{13}$ | 3 | 1 | 1 | 9 | 1 | 1 | 2 | 3 | 7 | 5 | 5 | 2.54 | 16.535 |
| $U_{14}$ | 0.143 | 0.111 | 0.111 | 1 | 0.125 | 0.111 | 0.125 | 0.143 | 0.333 | 0.2 | 0.2 | 0.178 | 1.162 |
| $U_{15}$ | 3 | 1 | 1 | 8 | 1 | 1 | 2 | 3 | 7 | 5 | 5 | 2.513 | 16.359 |
| $U_{16}$ | 3 | 1 | 1 | 9 | 1 | 1 | 2 | 3 | 7 | 5 | 5 | 2.54 | 16.535 |
| $U_{17}$ | 2 | 0.5 | 0.5 | 8 | 0.5 | 0.5 | 1 | 2 | 6 | 4 | 4 | 1.613 | 10.501 |
| $U_{18}$ | 1 | 0.333 | 0.333 | 7 | 0.333 | 0.333 | 0.5 | 1 | 5 | 3 | 3 | 1.062 | 6.917 |
| $U_{19}$ | 0.2 | 0.143 | 0.143 | 3 | 0.143 | 0.143 | 0.167 | 0.2 | 1 | 0.333 | 0.333 | 0.283 | 1.841 |
| $U_{110}$ | 0.333 | 0.2 | 0.2 | 5 | 0.2 | 0.2 | 0.25 | 0.333 | 3 | 1 | 1 | 0.514 | 3.349 |
| $U_{111}$ | 0.333 | 0.2 | 0.2 | 5 | 0.2 | 0.2 | 0.25 | 0.333 | 3 | 1 | 1 | 0.514 | 3.349 |

$\lambda_{max}$ = 11.414, CI = 0.041, RI = 1.514, CR = 0.027 < 0.10.

**Table 8.** Judgment matrix of SZSOJ's second-level evaluation factors ($U_2$).

| Factor | $U_{21}$ | $U_{22}$ | $U_{23}$ | $U_{24}$ | $U_{25}$ | Eigenvector | Weight (%) |
|---|---|---|---|---|---|---|---|
| $U_{21}$ | 1 | 7 | 0.333 | 3 | 3 | 1.838 | 24.277 |
| $U_{22}$ | 0.143 | 1 | 0.111 | 0.2 | 0.333 | 0.254 | 3.355 |
| $U_{23}$ | 3 | 9 | 1 | 5 | 7 | 3.936 | 51.98 |
| $U_{24}$ | 0.333 | 5 | 0.2 | 1 | 3 | 1 | 13.205 |
| $U_{25}$ | 0.333 | 3 | 0.143 | 0.333 | 1 | 0.544 | 7.183 |

$\lambda_{max}$ = 5.231, CI = 0.058, RI = 1.11, CR = 0.052 < 0.10.

**Table 9.** Judgment matrix of SZSOJ's second-level evaluation factors ($U_3$).

| Factor | $U_{31}$ | $U_{32}$ | $U_{33}$ | $U_{34}$ | $U_{35}$ | $U_{36}$ | $U_{37}$ | Eigenvector | Weight (%) |
|---|---|---|---|---|---|---|---|---|---|
| $U_{31}$ | 1 | 0.333 | 0.333 | 1 | 0.2 | 0.167 | 4 | 0.548 | 5.38 |
| $U_{32}$ | 3 | 1 | 1 | 3 | 0.333 | 0.25 | 6 | 1.24 | 12.174 |
| $U_{33}$ | 3 | 1 | 1 | 3 | 0.333 | 0.25 | 6 | 1.24 | 12.174 |
| $U_{34}$ | 1 | 0.333 | 0.333 | 1 | 0.2 | 0.167 | 4 | 0.548 | 5.38 |
| $U_{35}$ | 5 | 3 | 3 | 5 | 1 | 0.5 | 8 | 2.643 | 25.95 |
| $U_{36}$ | 6 | 4 | 4 | 6 | 2 | 1 | 9 | 3.747 | 36.793 |
| $U_{37}$ | 0.25 | 0.167 | 0.167 | 0.25 | 0.125 | 0.111 | 1 | 0.219 | 2.15 |

$\lambda_{max}$ = 7.277, CI = 0.046, RI = 1.341, CR = 0.034 < 0.10

**Table 10.** Judgment matrix of SZSOJ's second-level evaluation factors ($U_4$).

| Factor | $U_{41}$ | $U_{42}$ | $U_{43}$ | Eigenvector | Weight (%) |
|---|---|---|---|---|---|
| $U_{41}$ | 1 | 7 | 0.333 | 1.326 | 28.974 |
| $U_{42}$ | 0.143 | 1 | 0.111 | 0.251 | 5.49 |
| $U_{43}$ | 3 | 9 | 1 | 3 | 65.536 |

$\lambda_{max}$ = 3.08, CI = 0.040, RI = 0.525, CR = 0.076 < 0.10.

The consistency test was performed on all the results, which showed that the weight set obtained through AHP is valid and reasonable.

The AHP analysis yielded varying weight values for each item, highlighting differences in their relative importance. For instance, $U_3$ (Functions within the zoo) in the first-level catalog had a weight value of 3.819%. In comparison, $U_{11}$ (Plants' QR code information) and $U_{18}$ (Animal education science videos QR code information) had a 6.917% weighting in the second-level catalog. In contrast, $U_{14}$ (Questionnaire research QR code information) had a weight value of only 1.162%. Similarly, $U_{19}$ (Regional activities' QR code information) had a 1.814% weighting, and $U_{110}$ (Animal education science live QR

code information) and $U_{111}$ (Animal protection organization QR code information) had a combined weight of 3.349%. The weight of $U_{22}$ (Environmental sensors) was 3.355%, and that of $U_{25}$ (Ecological energy use information) was 7.183%. On the other hand, $U_{31}$ (Free WIFI) and $U_{34}$ (Animal state observation) had weights of 5.38%, while $U_{37}$ (Smart souvenir vending (photos)) had a weight of 2.15%, and $U_{42}$ (Tourism SNS) had a weight of 5.49%. Interestingly, these weights were lower than expected, suggesting that visitors or citizens may not necessarily share the same expectations as researchers or designers regarding the envisioned smart features.

### 3.2. Results of FCEM

The exact values for each second-level evaluation factor of the questionnaire can be found in Tables 11–14.

**Table 11.** Results of the second-level questionnaire (QR code information function).

| Factor | Score | | | | |
|---|---|---|---|---|---|
| | 1 | 2 | 3 | 4 | 5 |
| Plant QR code information | 9.40% | 13.68% | 20.51% | 24.79% | 31.62% |
| Animal QR code information | 7.69% | 11.97% | 21.37% | 24.79% | 34.19% |
| Media report QR code information | 7.69% | 11.97% | 15.38% | 34.19% | 30.77% |
| Questionnaire research QR code information | 12.82% | 20.51% | 18.80% | 23.93% | 23.93% |
| Artwork QR code information | 5.98% | 13.68% | 15.38% | 35.90% | 29.06% |
| Academic research QR code information | 8.55% | 7.69% | 22.22% | 29.91% | 31.62% |
| Event QR code information | 5.98% | 12.82% | 20.51% | 31.62% | 29.06% |
| Animal education science videos QR code information | 7.69% | 13.68% | 22.22% | 24.79% | 31.62% |
| Regional activities QR code information | 7.69% | 15.38% | 26.50% | 31.62% | 18.80% |
| Animal education science live QR code information | 11.11% | 11.97% | 19.66% | 33.33% | 23.93% |
| Animal protection organization QR code information | 11.11% | 12.82% | 19.66% | 29.91% | 26.50% |

**Table 12.** Results of the second-level questionnaire (Ecology System).

| Factor | Score | | | | |
|---|---|---|---|---|---|
| | 1 | 2 | 3 | 4 | 5 |
| Ecological cycle systems | 5.13% | 14.53% | 19.66% | 35.04% | 25.64% |
| Environmental sensors | 5.98% | 15.38% | 26.50% | 31.62% | 20.51% |
| Automatic watering | 5.98% | 9.40% | 21.37% | 36.75% | 26.50% |
| Eco-energy (solar power) | 9.40% | 15.38% | 14.53% | 29.91% | 30.77% |
| Ecological energy use information | 9.40% | 12.82% | 22.22% | 29.06% | 26.50% |

**Table 13.** Results of the second-level questionnaire (Functions within the zoo).

| Factor | Score | | | | |
|---|---|---|---|---|---|
| | 1 | 2 | 3 | 4 | 5 |
| Free WIFI | 9.40% | 11.97% | 17.09% | 29.91% | 31.62% |
| Electronic ticketing system | 8.55% | 9.40% | 18.80% | 29.91% | 33.33% |
| Interactive animal education | 6.84% | 11.11% | 22.22% | 24.79% | 35.04% |
| Animal state observation | 11.11% | 12.82% | 16.24% | 25.64% | 34.19% |
| Animal status detection (camera) | 5.98% | 11.97% | 17.95% | 28.21% | 35.90% |
| Electronic information screen | 4.27% | 12.82% | 15.38% | 33.33% | 34.19% |
| Smart souvenir vending (photos) | 7.69% | 12.82% | 25.64% | 30.77% | 23.08% |

**Table 14.** Results of the second-level questionnaire (Official website function).

| Factor | Score | | | | |
|---|---|---|---|---|---|
| | **1** | **2** | **3** | **4** | **5** |
| Official website function | 7.69% | 11.97% | 19.66% | 28.21% | 32.48% |
| Tourism SNS | 14.53% | 6.84% | 23.08% | 23.08% | 32.48% |
| Digital map | 6.84% | 11.97% | 18.80% | 28.21% | 34.19% |

The weight values for the first-level Item Set A and the second-level Item Set $W_m$, calculated through the AHP method, are presented below:

$$A = \begin{bmatrix} 0.5119 & 0.1175 & 0.0382 & 0.3324 \end{bmatrix}$$

$$W_1 = \begin{bmatrix} 0.0692 & 0.1654 & 0.1654 & 0.0116 & 0.1636 & 0.1654 & 0.1050 & 0.0692 & 0.0184 & 0.0335 & 0.0335 \end{bmatrix}$$

$$W_2 = \begin{bmatrix} 0.2428 & 0.0336 & 0.5198 & 0.1321 & 0.0718 \end{bmatrix}$$

$$W_3 = \begin{bmatrix} 0.0538 & 0.1217 & 0.1217 & 0.0538 & 0.2598 & 0.3679 & 0.0215 \end{bmatrix}$$

$$W_4 = \begin{bmatrix} 0.2897 & 0.0549 & 0.6554 \end{bmatrix}$$

Based on the membership degree of the Item Set $R_m$, the following can be constructed:

$$R_1 = \begin{bmatrix} 0.25 & 0.35 & 0.20 & 0.12 & 0.09 \\ 0.26 & 0.27 & 0.20 & 0.14 & 0.14 \\ 0.26 & 0.21 & 0.23 & 0.16 & 0.15 \\ 0.26 & 0.27 & 0.20 & 0.12 & 0.15 \\ 0.22 & 0.33 & 0.21 & 0.14 & 0.10 \\ 0.26 & 0.24 & 0.21 & 0.08 & 0.21 \\ 0.22 & 0.32 & 0.15 & 0.15 & 0.16 \\ 0.23 & 0.24 & 0.26 & 0.15 & 0.13 \\ 0.21 & 0.33 & 0.21 & 0.09 & 0.15 \\ 0.26 & 0.28 & 0.15 & 0.20 & 0.12 \\ 0.20 & 0.38 & 0.20 & 0.11 & 0.12 \end{bmatrix}$$

$$R_2 = \begin{bmatrix} 0.22 & 0.26 & 0.26 & 0.10 & 0.15 \\ 0.23 & 0.28 & 0.20 & 0.15 & 0.15 \\ 0.25 & 0.26 & 0.22 & 0.10 & 0.17 \\ 0.26 & 0.30 & 0.21 & 0.09 & 0.14 \\ 0.28 & 0.27 & 0.18 & 0.14 & 0.13 \end{bmatrix}$$

$$R_3 = \begin{bmatrix} 0.32 & 0.29 & 0.15 & 0.14 & 0.09 \\ 0.32 & 0.30 & 0.20 & 0.13 & 0.05 \\ 0.28 & 0.21 & 0.27 & 0.13 & 0.11 \\ 0.32 & 0.26 & 0.21 & 0.12 & 0.09 \\ 0.29 & 0.25 & 0.20 & 0.13 & 0.14 \\ 0.34 & 0.22 & 0.25 & 0.12 & 0.07 \\ 0.20 & 0.33 & 0.23 & 0.10 & 0.14 \end{bmatrix}$$

$$R_4 = \begin{bmatrix} 0.31 & 0.33 & 0.14 & 0.11 & 0.11 \\ 0.24 & 0.28 & 0.21 & 0.17 & 0.10 \\ 0.32 & 0.22 & 0.21 & 0.15 & 0.10 \end{bmatrix}$$

Afterwards, the first-level fuzzy comprehensive evaluation result can be obtained by using the assessment matrix C and the corresponding weight vector A, as B = A × C.

$$C_1 = W_1 \times R_1$$

$$C_2 = W_2 \times R_2$$

$$C_3 = W_3 \times R_3$$

$$C_4 = W_4 \times R_4$$

$$B = A \times \begin{bmatrix} C_1 & C_2 & C_3 & C_4 \end{bmatrix}$$

$$B = \begin{bmatrix} 0.2694 & 0.2682 & 0.2034 & 0.1317 & 0.1298 \end{bmatrix}$$

The fuzzy comprehensive evaluation approach is commonly based on the maximum-membership degree principle to determine the results. Upon analysis of vector B, it is apparent that the membership-degree values corresponding to the ranking system's categories of "excellent", "good", "moderate", "fair", and "poor" are 0.2694, 0.2682, 0.2034, 0.1317, and 0.1298, respectively. Notably, the highest membership degree value of 0.2694 is attributed to the "excellent" category. Therefore, the SZSOJ evaluation score for Kyoto Zoo is calculated to be 0.2694, which reflects an "excellent" rating. This finding indicates that the intellectualization construction efforts of Kyoto Zoo are commendable, resulting in high levels of visitor satisfaction and agreement with the zoo's intellectualization initiatives.

### 3.3. Results of IPA

The arithmetic mean of all factor scores can be calculated using SPSS 21.0 software on the unprocessed data collected from the questionnaire, as tabulated in Tables 15 and 16. The generated IPA matrices are graphically depicted in Figures 2 and 3, which enable us to visually identify the key areas of concern and prioritize the corresponding demands.

**Table 15.** Means and rankings of items at the first level of classification.

| Index | Performance (P) | | Importance (I) | | Mean Difference (P-I) | t Value | p Value |
|---|---|---|---|---|---|---|---|
| | Average Value | Rank | Average Value | Rank | | | |
| QR code information function | 3.364 | 3 | 3.554 | 4 | −0.19 | −2.477 | 0.01 |
| Ecology system | 3.359 | 4 | 3.566 | 3 | −0.21 | −2.191 | 0.03 |
| Functions within the zoo | 3.540 | 1 | 3.667 | 1 | −0.13 | −1.852 | 0.07 |
| Official website function | 3.501 | 2 | 3.630 | 2 | −0.13 | −1.382 | 0.17 |

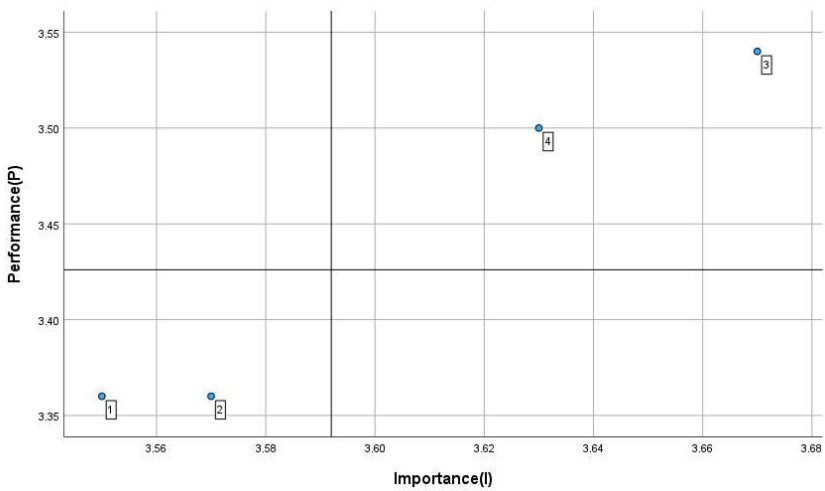

**Figure 3.** IPA matrix of items at the first level of classification. Note: 1—QR code information function; 2—Ecology system; 3—Functions within the zoo; 4—Official website function.

The IPA results present a stark contrast to the findings from the questionnaire, as illustrated in Figure 3. Notably, Functions within the zoo (categorized under the first quadrant) exhibited a significantly higher score than the mean values in both importance and expressiveness, thus emphasizing the need for its continuous sustenance. Similarly, the Official website function (also belonging to the first quadrant) scored higher than mean values in both importance and expressiveness, marking its significance. However, the QR code information function and Ecology system, both falling under the fourth quadrant, received below-average scores on both parameters, indicating their lower priority in the development program. Nevertheless, with sustained investment, these functions could be improved, and their recognition and value to visitors enhanced.

In summary, Functions within the zoo is the preeminent and efficacious aspect. In contrast, the QR code information function and Ecology system require additional investment to increase visitors' acknowledgment of their worth. The findings of this study underscore the need for continual refinement and enhancement of the smart features of the SZSOJ to sustain and elevate visitor satisfaction and engagement. As such, the integration of user-centered design principles and feedback mechanisms should be prioritized in developing and implementing smart features in zoo environments. By doing so, the SZSOJ can reinforce its position as a cutting-edge smart zoo and provide visitors with an exceptional and memorable experience.

**Table 16.** Means and rankings of items at the second level of classification.

| Index | Performance (P) | | Importance (I) | | Mean Difference (P-I) | t Value | p Value |
|---|---|---|---|---|---|---|---|
| | Average Value | Rank | Average Value | Rank | | | |
| Plant QR code information | 3.556 | 6 | 3.556 | 18 | 0.00 | 0.00 | 1.00 |
| Animal QR code information | 3.376 | 15 | 3.658 | 10 | −0.28 | −1.80 | 0.07 |
| Media report QR code information | 3.265 | 25 | 3.684 | 6 | −0.42 | −3.02 | 0.00 |
| Questionnaire research QR code information | 3.359 | 17 | 3.256 | 26 | 0.10 | 0.63 | 0.53 |
| Artwork QR code information | 3.436 | 10 | 3.684 | 6 | −0.25 | −1.58 | 0.12 |
| Academic research QR code information | 3.265 | 25 | 3.684 | 6 | −0.42 | −3.00 | 0.00 |
| Event QR code information | 3.291 | 23 | 3.650 | 12 | −0.36 | −2.57 | 0.01 |
| Animal education science videos QR code information | 3.299 | 22 | 3.590 | 15 | −0.29 | −1.97 | 0.05 |
| Regional activities QR code information | 3.376 | 15 | 3.385 | 25 | −0.01 | −0.06 | 0.95 |
| Animal education science live QR code information | 3.359 | 17 | 3.470 | 23 | −0.11 | −0.64 | 0.52 |
| Animal protection organization QR code information | 3.419 | 12 | 3.479 | 22 | −0.06 | −0.39 | 0.70 |
| Ecological cycle systems | 3.291 | 23 | 3.615 | 14 | −0.32 | −2.42 | 0.02 |
| Environmental sensors | 3.308 | 20 | 3.453 | 24 | −0.15 | −1.00 | 0.32 |
| Automatic watering | 3.308 | 20 | 3.684 | 6 | −0.38 | −2.62 | 0.01 |
| Eco-energy (solar power) | 3.444 | 8 | 3.573 | 17 | −0.13 | −0.86 | 0.39 |
| Ecological energy use information | 3.444 | 8 | 3.504 | 20 | −0.06 | −0.38 | 0.71 |
| Free WIFI | 3.615 | 3 | 3.624 | 13 | −0.01 | −0.06 | 0.95 |
| Electronic ticketing system | 3.718 | 1 | 3.701 | 4 | 0.02 | 0.14 | 0.89 |
| Interactive animal education | 3.419 | 12 | 3.701 | 4 | −0.28 | −2.01 | 0.05 |
| Animal state observation | 3.598 | 5 | 3.590 | 15 | 0.01 | 0.05 | 0.96 |
| Animal status detection (camera) | 3.427 | 11 | 3.761 | 2 | −0.33 | −2.26 | 0.03 |
| Electronic information screen | 3.650 | 2 | 3.803 | 1 | −0.15 | −1.14 | 0.26 |
| Smart souvenir vending (photos) | 3.350 | 19 | 3.487 | 21 | −0.14 | −0.87 | 0.39 |
| Official website function | 3.615 | 3 | 3.658 | 10 | −0.04 | −0.28 | 0.78 |
| Tourism SNS | 3.385 | 14 | 3.521 | 19 | −0.14 | −0.89 | 0.38 |
| Digital map | 3.504 | 7 | 3.709 | 3 | −0.21 | −1.56 | 0.12 |

Figure 4 provides clear evidence that the Animal status detection (camera) function is highly valued by visitors and, therefore, should be prioritized for continued devel-

opment and maintenance. However, the Electronic information screen, Ecological cycle systems, Animal education science live QR code information, Animal protection organization QR code information, and Artwork QR code information are less highly valued by visitors. They should therefore be given lower priority in future development efforts. Conversely, visitors have expressed an interest in Plant QR code information, indicating its potential as a feature that could be further developed. Overall, the majority of the features fall in or around the center of the graph, with some outliers in the fourth quadrant, suggesting the need for consistent development and maintenance efforts.

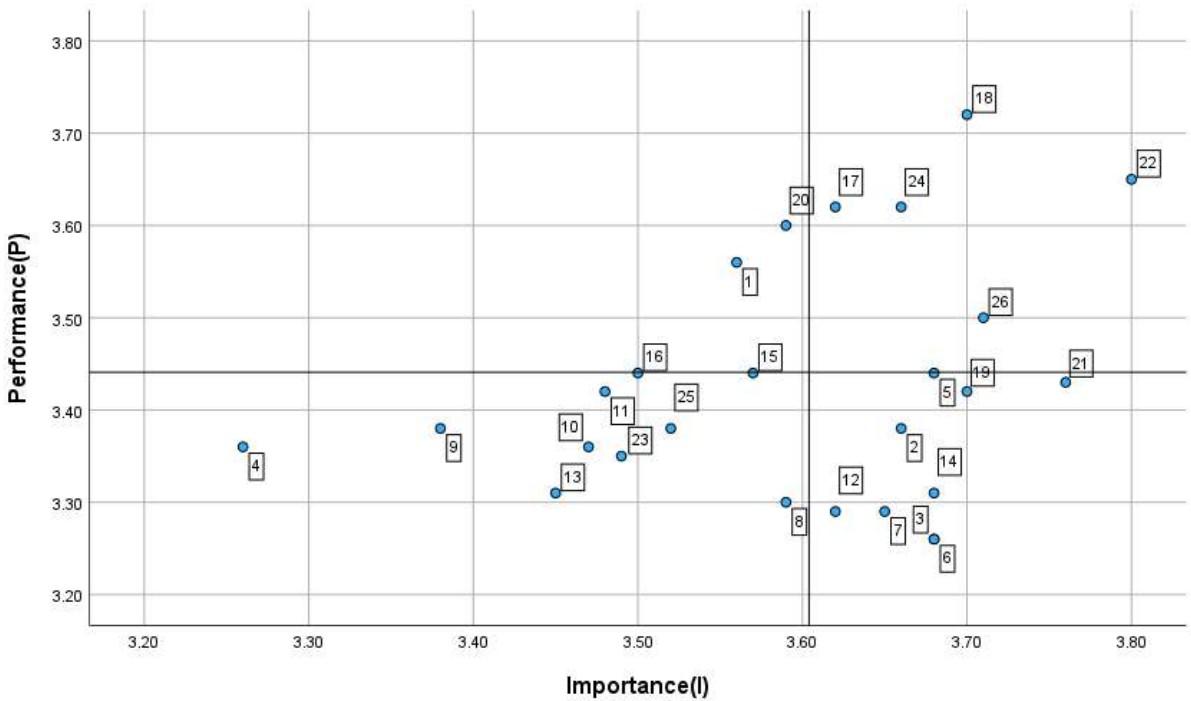

**Figure 4.** IPA matrix of items at the second level of classification. Note: 1—Plant QR code information; 2—Animal QR code information; 3—Media report QR code information; 4—Questionnaire research QR code information; 5—Artwork QR code information; 6—Academic research QR code information; 7—Event QR code information; 8—Animal education science videos QR code information; 9—Regional activities QR code information; 10—Animal education science live QR code information; 11—Animal protection organization QR code information; 12—Ecological cycle systems; 13—Environmental sensors; 14—Automatic watering; 15—Eco-energy (solar power); 16—Ecological energy uses information; 17—Free WIFI; 18—Electronic ticketing system; 19—Interactive animal education; 20—Animal state observation; 21—Animal status detection (camera); 22—Electronic information screen; 23—Smart souvenir vending (photos); 24—Official website function; 25. Tourism SNS; 26. Digital map.

*3.4. Results on Satisfaction of Zoo Visitors*

We harnessed the study to distill a singular gauge of zoo visitor contentment, reflecting their perceptions within the current context. Each score was multiplied by the corresponding item's satisfaction proportion, culminating in an averaged overall value harmonized with the weights of each first-level categorization, yielding a final satisfaction rating out of 5. The synthesis of different factors yielded a weighted mean satisfaction score in Kyoto Zoo of 3.43 (compared to Ueno Zoo's 2.70), affirming visitors' positive sentiments. Generally, a score greater than 3 indicates good satisfaction. This consolidated metric, aligned with scholarly practices, encapsulates smart features, sustainability, and visitor-centric amenities, reflecting the holistic zoo experience. This approach underscores methodological rigor, resonating with academic discourse, and deepens our understanding of smart zoos' impact on visitor satisfaction dynamics.

## 4. Discussion

### 4.1. Findings from the Questionnaire

The findings from the questionnaire survey conducted at Kyoto Zoo have yielded insightful results, with most items scoring similarly and possessing little disparity in terms of importance and expressiveness. However, some unexpected revelations emerged, such as the Functions within the zoo being ranked the least important among the four items in the first level of classification, exhibiting a significant value gap. In contrast, the QR code information function was surprisingly rated as the most important. Moreover, the questionnaire collection process and results differed from those of Ueno Zoo, and the following specific observations were identified:

1. Firstly, there was a marked difference between Kyoto Zoo and Ueno Zoo in terms of questionnaire awareness. The feedback from the questionnaire about Ueno Zoo revealed that many respondents needed to be made aware of the existence of some smart functions in the park if there were no accompanying photos. In contrast, clarity was sufficient for the completion of questionnaires at Kyoto Zoo, indicating a more thorough understanding of these smart functions among citizens. This may be attributed to the fact that the good promotion of smart features in Kyoto City's smart city project has fostered widespread acceptance and comprehension of smart functions among the populace [27], unlike in Ueno Zoo, where the importance and performance of many projects exhibit significant disparities.

2. Secondly, the present study examined and compared the feedback received from visitors at Kyoto and Ueno Zoos regarding the importance and performance of various smart functions. Interestingly, the results showed that there was a significant difference between the two zoos in the importance of Functions within the zoo. While this function was ranked the least important among the four items in the first level of classification in Kyoto Zoo, it was surprisingly ranked the most important function by respondents in the Ueno Zoo questionnaire. This may be due to the differing scale and positioning of the two zoos. Ueno Zoo, being a zoo with a large flow of people in the city center and many foreign visitors, may have visitors who pay more attention to offline interactive functions without the use of devices. In contrast, Kyoto Zoo, being a regional city zoo welcoming mostly resident visitors, may have visitors who expect newer and more innovative intelligent functions. Additionally, the respondents at Kyoto Zoo may have perceived Functions within the zoo as a basic feature that does not require much attention or specialness, as its project performance is similar to that of the city streets outside the park (e.g., the free Wi-Fi function at Kyoto Zoo uses the city Wi-Fi of Kyoto City). However, visitors to both zoos were found to value Official website functions highly, with visitors showing a strong demand for information about official releases. Moreover, the regional service nature of Kyoto Zoo may have contributed to the need for regional communication functions such as the QR code information function. These findings shed light on the different factors that may influence visitor perceptions and expectations of smart functions in zoos and highlight the need for zoos to carefully consider their unique visitor profiles when designing and implementing smart features.

3. Finally, we propose that the promotion of smart city projects in Kyoto City and the financial crisis of the past few years have raised awareness and expectations of smart cities, which may lead to higher average feedback scores on the importance scale in the future.

### 4.2. Findings from Analytical Calculations

The results of the FCEM analysis demonstrate that the intellectualization infrastructure of Kyoto Zoo is deemed "excellent" (with an FCEM evaluation score of 0.2694). This finding suggests that citizens can easily comprehend and appreciate the intellectualization features of the zoo. Although unexpected, this is a very positive outcome, as it indicates that Kyoto Zoo can effectively realize the intellectualization process within

the Smart City framework, making it more accessible and integrated into citizens' daily lives. Furthermore, in contrast to the FCEM result of Ueno Zoo, which received a "fair" score, the importance of the smart city background and system is more prominently manifested in the smart zoo concept [13]. This is due to the smaller scale of Kyoto Zoo and its amiable service style. Therefore, the public may prioritize practical features that have frequent daily uses over those that appear technologically advanced, akin to the higher happiness satisfaction reported in small towns compared to big cities.

The IPA analysis yielded results that differ significantly from the numerical importance ratings obtained from the questionnaire in the first classification level. We posit the following explanations:

1. The distinction arises from the questionnaire design, where importance is assessed solely at the first level of categorization. The respondents' direct voting on these first-level categories determines their importance, hinging on their judgment of the overarching functional categorization. In contrast, IPA generates an average value by incorporating all respondents' responses to second-level categorization items in the calculation. This approach is more specific and depends on each functional category's sub-item performance. The questionnaire's importance value directly stems from tallying first-level categorical items, while IPA calculates the mean of its second-level categorical items.

2. Overall satisfaction (derived from direct scoring of first-level categorical items in the questionnaire) may vary based on visitors' perceptions. For instance, the QR code information function, primarily focused on digital interaction, might prompt visitors to anticipate a comprehensive zoo intelligence. Conversely, "Functions within the zoo" is a broader category found in various Japanese zoos, making it challenging to associate directly with overall intelligence satisfaction. IPA's mean value for second-level category items differs in this aspect. Some first-level category items may exhibit relatively lower overall satisfaction scores but have sub-categories (e.g., "Animal Status Detection (camera)" within "Functions within the zoo") that garner high satisfaction. Consequently, these items receive higher values in IPA's mean value calculation.

3. The quantity of sub-items varies across each Level 1 categorical item. For instance, the first category, "QR code information function", encompasses 11 sub-items, whereas the fourth, "Official website function", includes only 3. This disparity in sub-item count could influence visitors' perceptions and expectations. The QR code information function, featuring numerous sub-items, might overwhelm visitors with its multitude of functions, possibly eliciting feelings of fatigue or numbness. Indeed, our subjective interviews revealed inquiries like, "Why doesn't the zoo consolidate all these functions into one platform?"

The QR code information function and the Ecology system require further development and refinement to increase public and visitor awareness of their significance in driving the park's sustainable growth.

The current strengths, weakness, opportunities, and threats of Kyoto Zoo are summarized in the SWOT chart in Figure 5.

### 4.3. Comparison and Recommendations for Ueno Zoo Based on the Impact of Kyoto Smart City

Regarding system classification, both Kyoto Zoo and Ueno Zoo are classified as shown in Figure 6.

First, we will examine both zoos in a combined weighted order. We ranked the weights of the smart items of Kyoto Zoo (including the first-quarter classification and the second-level classification) obtained from Tables 6–10 and compared them with the items from Ueno Zoo. The results of the weights of the first-level classified items and the second-level classified items for each ranking are shown in Figures 7 and 8.

Figures 7 and 8 show that the item weights in Kyoto Zoo exhibit a higher degree of differentiation than those in Ueno Zoo. The range between the maximum and minimum values in Kyoto Zoo is more pronounced. Furthermore, Figure 8 highlights that approxi-

mately 20 sub-items in Kyoto Zoo have weights below 20%, with 6 sub-items falling below 5%. Furthermore, there is a substantial disparity in the weights of the top four sub-items in Kyoto Zoo. In contrast, Ueno Zoo displays a relatively uniform distribution of item weights in the second classification level, resulting in a more balanced overall distribution. Interestingly, even the weights of the first three items in Ueno Zoo are identical.

The magnitude of weighting also mirrors the visitors' level of expectation. In the case of Kyoto Zoo and Ueno Zoo, some of the programs with a high weighting (which, on the other hand, is interpreted as programs that visitors strongly anticipate) did not perform well and, therefore, did not end up in Quadrant 1 or even Quadrant 4 of the IPA results, which indicates that the programs developed by the zoos sometimes do not correspond to the actual needs of the visitors.

Secondly, we need to compare the two zoos' respective performances at the current stage. Concerning the overall FCEM results (Kyoto: excellent, Ueno: fair), Kyoto Zoo aligns better with visitors' perceived needs for smart features. Additionally, in terms of the single satisfaction value (Kyoto: 3.43, Ueno: 2.70), Kyoto Zoo outperforms Ueno Zoo. In other words, based on the current state of development, Kyoto Zoo's smart projects are better suited to the needs of local tourists and the collaborative development required for a zoo. Despite Ueno Zoo having more construction funds and a larger scale, visitor feedback on its current performance prompts considerations about whether more advanced intellectualization is always better, or whether finding smart projects suitable for the public represents a more favorable development concept.

Thirdly, we need to compare them regarding the overall project categorization framework. As no unified smart management platform exists, Kyoto Zoo cannot be classified using the same criteria as Ueno Zoo at the first level. However, it can be classified based on the direction of functional development. Currently, Kyoto Zoo has fewer first-level classifications due to the lack of smartphone applications. However, it has a strong QR code information function, classified as a first-level item. The Ecology system is also a primary development direction at Kyoto Zoo and a first-level item.

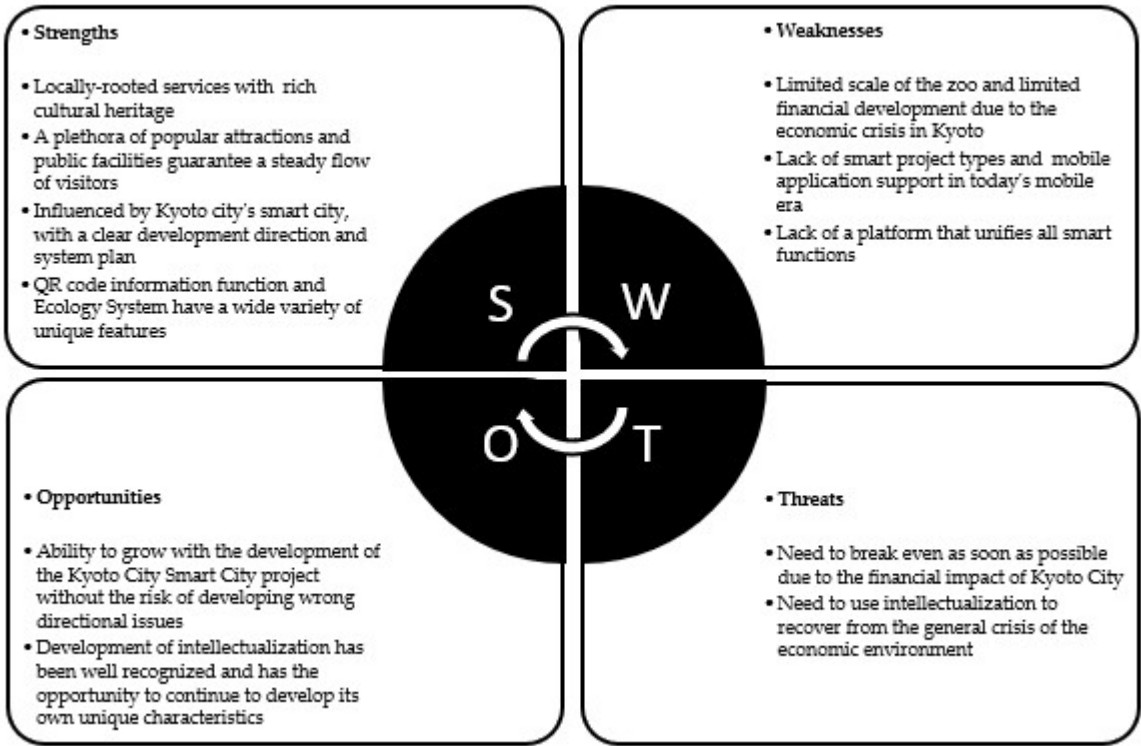

**Figure 5.** SWOT of the current state of Kyoto Zoo.

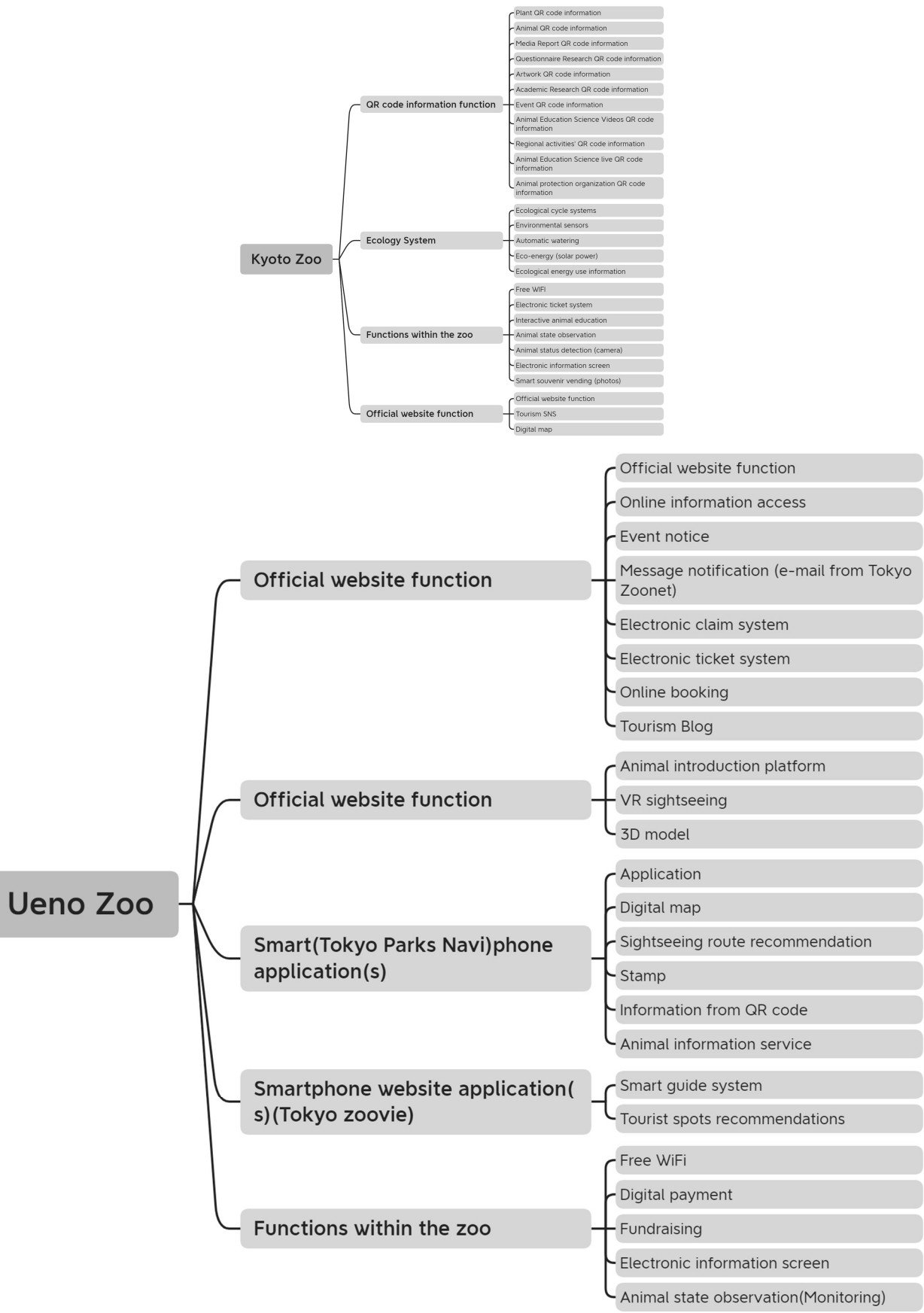

**Figure 6.** The framework system of the current state of Ueno Zoo and Kyoto Zoo.

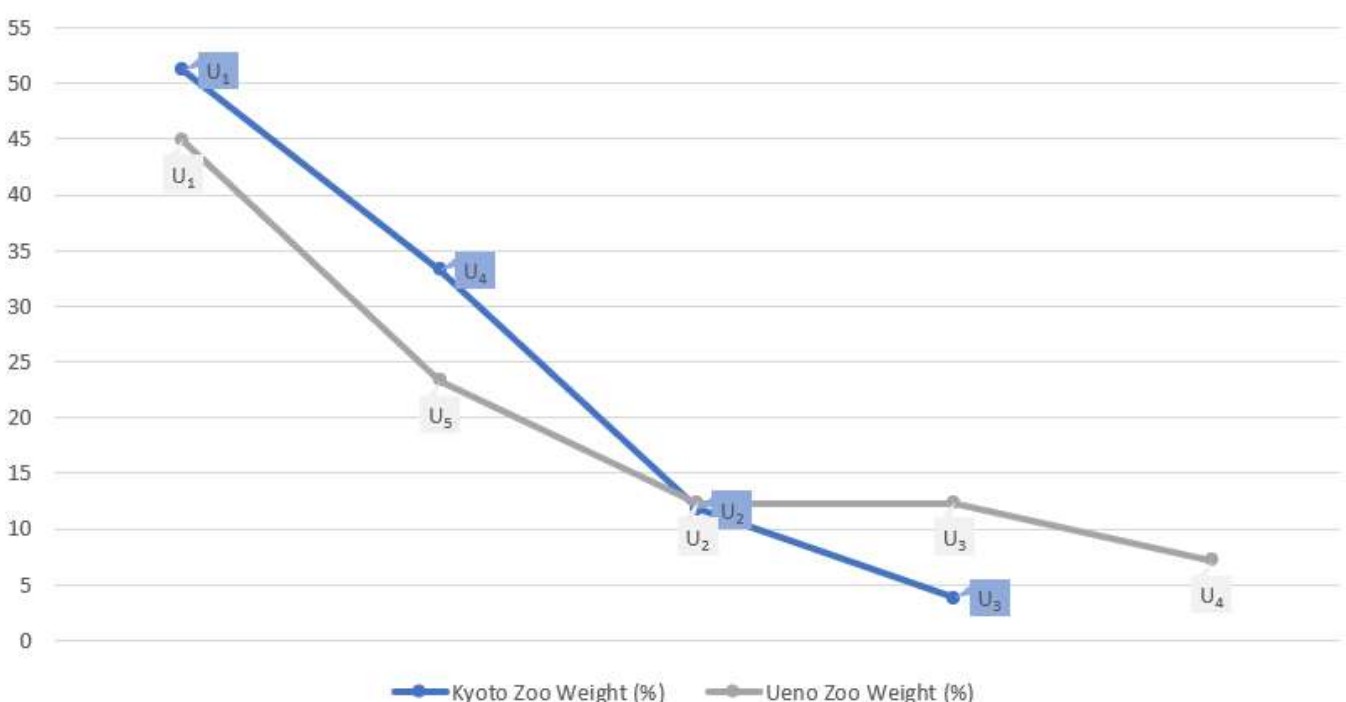

**Figure 7.** Project weighting of the first level of classification for Ueno Zoo and Kyoto Zoo.

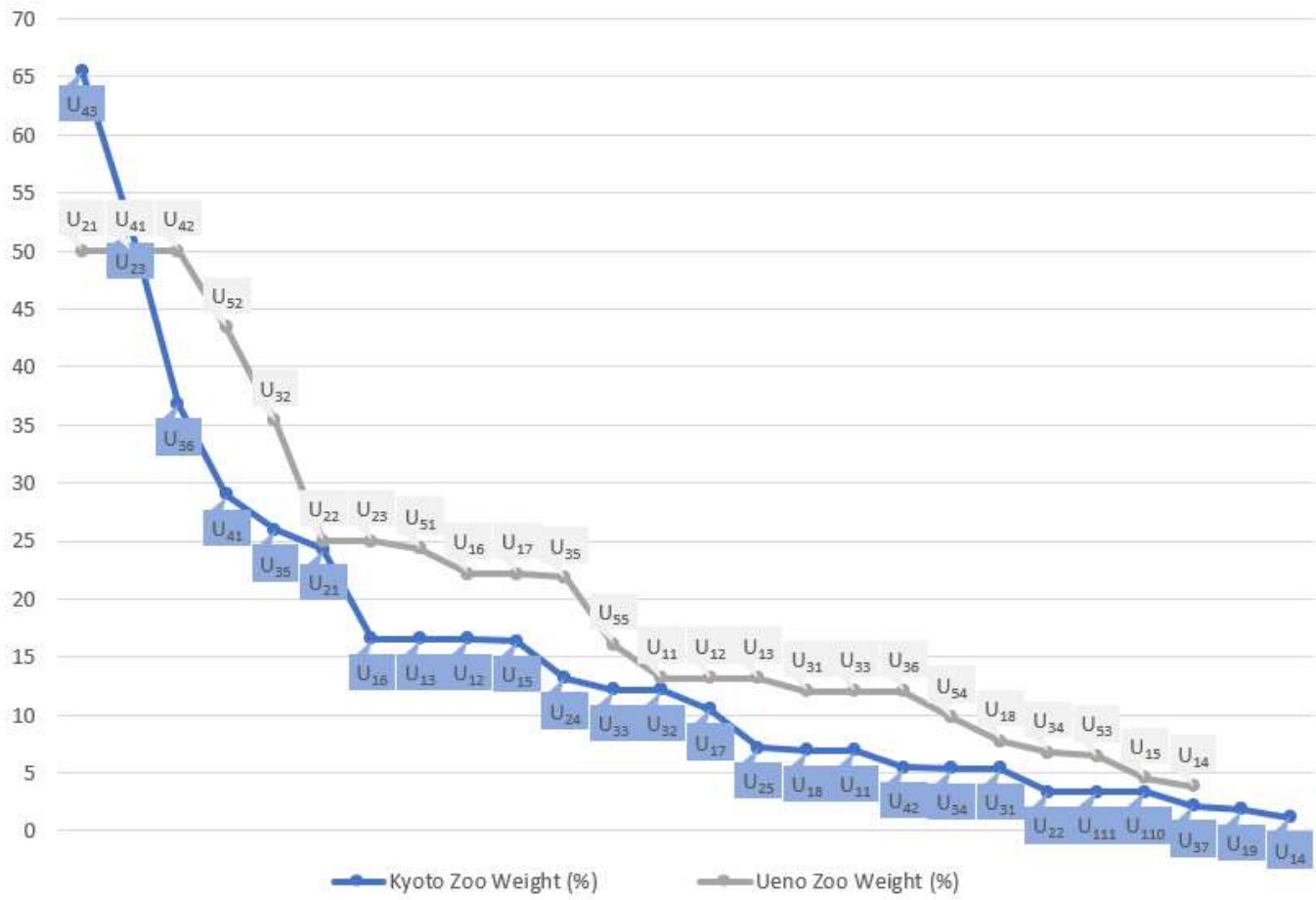

**Figure 8.** Project weighting of the second level of classification for Ueno Zoo and Kyoto Zoo.

On the other hand, the Official website functions and Functions within the zoo have fewer secondary classification sub-projects. Although Kyoto Zoo scored well in the overall IPA score, considering the limited classification coverage, we suggest that Kyoto Zoo should increase its coverage in future planning. Therefore, we recommend that Kyoto Zoo increase its classification coverage for better development.

Although Kyoto Zoo's current intellectualization only focuses on its ecology system, the fact that it is already part of the smart city development plan and has proposed regional smart equipment is an encouraging sign. It is also promising that the city's pre-existing smart facilities, such as the smart traffic system, can be integrated with the zoo's intellectualization. With the ongoing development of the city's smart infrastructure, including the use of big data, human flow monitoring data, smart streetlights, and AI cameras, Kyoto Zoo has the potential to significantly enhance its smart capabilities. We strongly recommend that Kyoto Zoo take these opportunities into consideration when developing its future smart plans and categories. Doing so will allow the zoo to fully leverage its position within the smart city and take its intellectualization process to the next level.

## 5. Conclusions

The primary objective of this study is to ascertain the level of intellectualization in Japanese zoos by utilizing the FCEM analysis method while determining weights using the AHP. Additionally, this study aims to identify the current strengths and weaknesses of smart function developments in zoos through IPA and explore the prospects of such developments. At the same time, we compared Kyoto Zoo with Ueno Zoo to see the difference in intellectualization achievements in different contexts in terms of data and systems. Furthermore, this study aims to investigate the differences between Kyoto Zoo under the smart city system and a conventional smart zoo. As the concept of smart zoos is relatively novel, particularly in Japan, where smart cities are still in their developmental stages, we seek to refine objective system research methods to assess the intellectualization process more objectively, ultimately aiding zoos in Japan and around the world to become smarter. Our study results can be compared with current policies and be used to guide future developments in the field.

However, it is important to note that there are some limitations that can inform future research. Firstly, the selection of the smart project was influenced by certain characteristics unique to Kyoto Zoo, such as its difference in service orientation and smart project offerings, which made it difficult to compare with Ueno Zoo using the same criteria. Instead, we had to rely on feedback from service recipients to analyze questionnaire responses. We plan to conduct a comparative study once a unified standard for smart zoos is established in Japan again. Secondly, due to geographical constraints, Kyoto Zoo's lack of a cell phone applications and a smart platform for unified management may have limited public perception of smart functions. These limitations highlight the need for more comprehensive and standardized evaluations of smart zoos in the future.

In addition, future studies can explore more advanced and innovative smart functions in zoos, including advanced technologies like AI, the IoT, and big data analysis [28]. Moreover, as the concept of a smart city continues to evolve, it will be important to compare the development of smart zoos with other traditional parks in the city to better understand the impact of smart technology on the overall tourism industry. This can be achieved through AHP for decision making and can expand the scope of smart research beyond individual zoo analysis.

**Supplementary Materials:** The following supporting information can be downloaded at: https://www.mdpi.com/article/10.3390/land12091747/s1, Supplementary File S1: The following is the supplementary data related to this article.

**Author Contributions:** Conceptualization, Y.L. and R.S.; methodology, Y.L. and R.S.; software, Y.L.; validation, Y.L.; formal analysis, Y.L.; investigation, Y.L.; resources, Y.L.; data curation, Y.L. and Z.X.; writing—original draft preparation, Y.L. and S.L.; writing—review and editing, Y.L. and R.S.;

visualization, Y.L.; supervision, Y.L. and R.S.; project administration, Y.L.; funding acquisition, R.Y. All authors have read and agreed to the published version of the manuscript.

**Funding:** This research was supported by JST SPRING (No. JPMJSP2109, Japan).

**Data Availability Statement:** Not applicable.

**Conflicts of Interest:** The authors declare no conflict of interest.

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
