# Peer review of "Unveiling the Role of Zoos in Smart Cities: A Quantitative Analysis of the Degree of Smartness in Kyoto City Zoo"

_land, doi:10.3390/land12091747_

Round 1

Reviewer 1 Report

This research is of considerable interest in the direction of intellectualizing the sphere of life in cities and achieving the goals of their sustainable development. This positive result indicates that the Kyoto Zoo can effectively embody the process of intellectualization in the Concept of a "Smart City", making it more accessible and integrated into the daily lives of citizens.

as a comment, we can recommend the authors of the article:

1. Give explanations on the complex evaluation indicators U= U1...U4. Lines 280-285.

2. Come to a single indicator reflecting the satisfaction of zoo visitors as a result of the study.

Author Response

Thank you very much for your review and feedback. We have made the necessary revisions to the article and have provided explanations for the complex evaluation indicators U = U1...U4. Additionally, we have added a new chapter(3.4) to come up with a single indicator reflecting the satisfaction of zoo visitors as a result of the study.

Once again, we sincerely appreciate your valuable input, which has greatly benefited our paper.

Reviewer 2 Report

Although the objective is interesting, I think the paper is not original considering your previous mentioned paper. The theoretical framework, the methodology and the sample (students surveyed) are almost the same. The only difference is the case study. If you want to be original, you should focus on the comparison between the cases.

I recommend ordering the weights obtained in tables 6-10 in order of importance from the highest weights to the lowest.

In discussion, I think there is a contradiction in this sentence  (lines 497 …) “The findings from the questionnaire survey conducted at Kyoto Zoo have yielded  insightful results, with most items scoring similarly and possessing little disparity in terms of importance and expressiveness. However, some unexpected revelations emerged with IPA results.... Looking at the results and figure, items score differently and with disparity, that’s  why functions of the zoo scores higher than the others. This misunderstanding is reinforced at the end when you say:  “The IPA analysis yielded results that differ significantly from the numerical importance ratings obtained from the questionnaire in the first classification level”

This result is confusing, and although the authors discussed it, I think it deserves more explanation. This result does not take place in the previous case study. Besides, How do you complement the findings from questionnaire with those from the IPA?

Only 9 out of 25 references are articles in indexed journals. I think you can add more references in the subject from papers:

It would be useful to add references of recent papers using IPA, for instance:

Ban, O.-I.; Faur, M.-E.; Botezat, E.-A.; Ștefănescu, F.; Gonczi, J. An IPA Approach towards Including Citizens’ Perceptions into Strategic Decisions for Smart Cities in Romania. Sustainability 2022, 14, 13294. https://doi.org/10.3390/su142013294

And also for AHP related with smart cities:

https://www.mdpi.com/2227-7390/9/4/304

These references or others you find appropiate can improve the discussion section.

This sentence is written twice (lines 67-68) (line 63-64): aligning with the core principles and objectives of these technologically advanced urban environments. Delete one

You need to conserve anonymity of the paper, without saying “our previous paper Impact of Intellectualization of a Zoo through a FCEM- 176 AHP and IPA Approach”

Author Response

Thank you very much for your thorough review and valuable feedback. We have made the following modifications to the article:
1: We have added more comparisons between the cases in the comparison section of the discussion chapter (4.3). We agree with you and thank you for your comments about ordering the weights obtained in tables 6-10 in order of importance. We very much agree with and appreciate your comments on ordering the weights obtained in tables 6-10 in order of importance from the highest weights to the lowest, and have made this section into a chart and discussion of the comparison, as well as adding some comparisons between the current performance of the two zoos. This is combined with the original comparison of zoo intelligence frameworks to increase the hierarchy of comparisons and to focus on case-by-case comparisons wherever possible.
2: We have explained the confusing points (4.2) and included some additional explanations on the questionnaire and IPA results in three points, which are also discussed later in the discussion section.
3: We have added three references on methodology and smart city research. (references 22/24/26)
4: We deleted duplicate words.
5: We modified the citations' narrative to maintain the paper's anonymity.
Finally, thank you again for your review and comments. Many of the issues you pointed out are ones we have struggled with before, so we agree and thank you for your solutions.

Round 2

Reviewer 2 Report

I recommend to add a sentence in the abstract about the comparison with the Ueno Zoo. I also suggest you to add that information in the conclusion. 

Author Response

We have added a sentence in the Abstract and Conclusion part  about the comparison with the Ueno Zoo.

Very thanks for the recommendation.
